# Training A Multi-stage Deep Classifier with Feedback Signals

## Abstract

Multi-Stage Classifier (MSC) - several classifiers work sequentially in an arranged order and classification decision is partially made at each step - is widely used in industrial applications for various resource limitation reasons. The classifiers of a multi-stage process are usually Neural Network (NN) models trained independently or in their inference order without considering the signals from the latter stages. Aimed at two-stage binary classification process, the most common type of MSC, we propose a novel training framework, named Feedback Training. The classifiers are trained in an order reverse to their actual working order, and the classifier at the later stage is used to guide the training of initial-stage classifier via a sample weighting method. We experimentally show the efficacy of our proposed approach, and its great superiority under the scenario of few-shot training.

## 1 Introduction

The state-of-the-art deep neural networks have equipped a various of applications with much better quality, especially the emergence of BertDevlin et al. (2018), a TransformerVaswani et al. (2017)-based pre-training language model, and a series of its derivatives Brown et al. (2020); Lan et al. (2019). Their great success is mainly due to its scalability to encode large-scale data and to maneuver billions of model parameters. However, it is rather difficult to deploy them to real-time products such as Fraud Detection Senator et al. (1995); Kirkland et al. (1999), Search and Recommendation systems Covington et al. (2016); Ren et al. (2021), and many mobile applications, not only because of the high computational complexity but also the large memory requirements.

Several techniques are developed to make the trade-off between performance and model scale. Knowledge Distillation (KD) Hinton et al. (2015); Sanh et al. (2019) is the most empirically successful approach used to transfer the knowledge learnt from a heavy Teacher to a more light-weight and faster Student. Besides, Pruning Han et al. (2015); Frankle & Carbin (2018) and Quantization Han et al. (2015); Chen et al. (2020) further compress deep models even smaller. In many practical situations, however, we need super tiny models to meet the demanding memory and latency requirements, which would inevitably suffer serious performance degradation.

From another perspective, multi-stage classification system Trapeznikov et al. (2012) is widely used to reduce the opportunity of calling the deep and cumbersome models by filtering out some or even most of the input samples using simpler and faster models trained with limited data and easier features. In a multi-stage system, light-weight models such as SVM, Logistic Regression or k-Nearest Neighbors are used as earlier stage classifiers, classifying the samples (usually relatively easier negative ones) based on simple or easily accessible features, and leaving indeterminate ones for later. Models of later stages need to be heavier to deal with harder samples as well as more complex and costly features. A two-stage working mechanism is simply shown in Figure 1(a).

In several practical multi-stage applications, as shown in Figure 1(b) and (c), classifiers in different stages are trained independently or sequentially without considering the relationships among them Isler et al. (2019); Kruthika et al. (2019).

To build tighter connections between classifiers in a multi-stage system for better collaboration, most exist methods Mendes et al. (2020); Qi et al. (2019); Sabokrou et al. (2017); Zeng et al. (2013) jointly optimize the multi-stage classifiers in a way like cascade, allowing the contextual information to transfer from earlier stages to later. However, most of them primarily consider classification

accuracy rather than latency, and therefore would not make the classification decisions until the final stage.

In this paper, we consider to further explore how to forge closer connections between classifiers in a two-stage classification problem. We propose a novel training framework, Feedback Training, where the whole decision-making pipeline is consisted of two classifiers, a extremely lightweight Pre-classifier followed by a relatively heavier Main-classifier. Different from existing methods, these two models are trained in the reverse order of inference, that is, the first-stage model would be trained under the guidance of the second-stage one through a sample weighting method. The capacity of Pre-classifier is more effectively explored by considering the learning results of Main-classifier.

Our contributions can be summarized threefold:

1) We propose a novel training framework for two-stage classification applications.

2) We discuss a sample weighting method that assists Pre-classifier to learn according to its preference.

3) We verify our approach on two data sets that it outperforms baseline models significantly and shows greater superiority under few-shot scenarios.

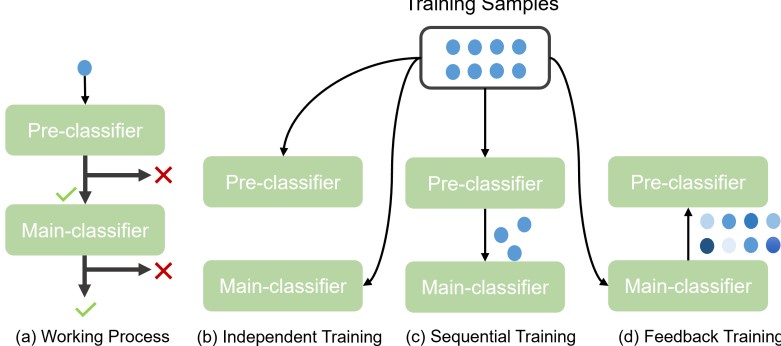

Figure 1: Working process and different training strategies for two-stage classifiers. We use the terms Pre-classifier and Main-classifier to denote the classifiers working at $1^{st}$ and $2^{nd}$ stages. (a) The working process of a two-stage classifier: only samples passed Pre-classifier would be fed into heavier Main-classifier for final decision, otherwise would be judged as negative without calling Main-classifier. (b) Independent Training: all classifiers are trained independently without considering the training results of each other. (c) Sequential Training: Classifiers are trained in their working order. Only samples passed Pre-classifier would be fed into Main-classifier for training. (d) Feedback Training: Classifiers are trained in their reverse order of inference. Pre-classifier assigns different attention to different samples based on the training result of Main-classifier and the proposed sample weighting approach.

## 2 PRELIMINARIES

We consider a binary classification problem with the training set $\mathcal{D} = \{(x_1, y_1), \ldots, (x_n, y_n)\}$, where $x_i$ denotes $i^{th}$ observed training sample paired with its label $y_i \in \{0, 1\}$. In a m-stage process, there are a series of predictive functions, $\mathcal{F} = \{f_{\theta_j}(\cdot)|j = 1, 2, ..., m\}$ working in the given order. The $j^{th}$ predictive function is parameterized by $\theta_j$. The classification decision is partially made in each step. One popular design is to filter out negative samples as many as possible in earlier stages and leaving the positive ones to the end for final decision. When classifiers are trained independently without considering the others, each one is trained by optimizing the basic Cross-Entropy loss as in Eq.1:

$$\mathcal{L}(f_{\theta_j}(\cdot)) = -\frac{1}{n}\sum_{i=1}^{n}\mathcal{L}_{\mathcal{CE}}(y_i, f_{\theta_j}(x_i)) \tag{1}$$

$$where\ \mathcal{L}_{\mathcal{CE}}(y, \hat{y}) = y \cdot log\hat{y} + (1 - y) \cdot log(1 - \hat{y}) \tag{2}$$

We also define a set of thresholds $\mathcal{T} = \{th_j | j = 1, 2, ..., m\}$ for each classifier. During inference, samples whose predictive scores are lower than corresponding thresholds would be judged as negative, and would not be sent to the next stage.

In sequential training, only the samples that pass all the previous classifiers are fed into the next stage for training. We formally define an indicator function $\mathbb{I}_i^j = 1\{\forall_{k<j} Score_i^k > th_k\}$ to denote whether $x_i$ would be used to train the $j^{th}$ classifier. $Score_i^k$ is the predictive score of $k^{th}$ classifier for $x_i$. Hence the total samples used to train $f_{\theta_j}(\cdot)$ is $\widetilde{n}_j = \sum_{i=1}^n \mathbb{I}_i^j$. Then we optimize Eq.3:

$$\mathcal{L}(f_{\theta_j}(\cdot)) = -\frac{1}{\widetilde{n}_j} \sum_{i=1}^n \mathbb{I}_i^j \cdot \mathcal{L}_{\mathcal{CE}}(y_i, f_{\theta_j}(x_i)) \tag{3}$$

## 3 FEEDBACK LEARNING

### 3.1 BACKGROUND AND MOTIVATION

Classification problems are becoming easier, achieving high performance simply by fine-tuning those pre-made large-scale language models with enough high quality observed samples. However, serving those heavy models in practical applications is quite challenging due to their demanding requirements for memory and latency. Therefore, we design a muti-stage classifier for faster inference.

Aimed at binary classification problems, our decision system consists of two networks combined, as demonstrated in Figure 1(a). Pre-classifier is a light-weight Logistic Regression model running on all input and tries its best to identify positive samples and meanwhile filter out as many negative ones as possible. The Main-classifier is a Transformer-base heavier network. Samples passed first-stage are potentially positive ones and would be fed into Main-classifier for final judgement. Similar workflows are widely used practically. One common learning strategy is to straightforwardly train the Main-classifier only with samples passed Pre-classifier as shown in Figure 1(c). However, this method arises a dual funnel Mendes et al. (2020) issue where a great number of samples are filtered out by the initial stage and the Main-classifier trained on this bias data set would lose its generalization ability.

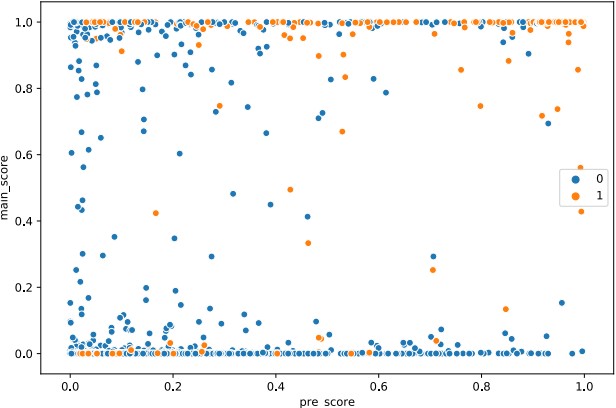

Figure 2: Scores predicted by Main-classifier and Pre-classifier.

If classifiers are trained independently (see Figure 1(b)), there would be an inconsistency problem shown in Figure 2. Some positive cases passed Pre-classifier may be rejected by Main-classifier (useless recalls of Pre-classifier), but some other positive cases rejected by Pre-classifier while they are highly scored by Main-classifier (potentially valuable recalls). If Pre-classifier rotates the scoring preference of these two kinds of cases, the overall performance would be improved. The same is true for negative samples. Therefore, modeling this preference is undoubtedly useful for the overall

performance, even without improving the accuracy of the classifiers. To this end, we propose the Feedback Training, a reversed order training framework, as described in the next section.

## 3.2 FEEDBACK TRAINING FRAMEWORK

**Main-classifier** In order to maintain the generalization performance of the Main-classifier, we train it independently in the first place on full training dataset with Cross-Entropy loss as shown in Eq.1.

**Pre-classifier** To better collaborate with the Main-classifier in a combined pipeline with its limited capacity, the Pre-classifier is trained with the following learning preferences:

- Pass positive samples and pay more attention to the samples could be identified by Main-classifier.

- Filter out negative samples and those could not be identified by Main-classifier (whether they are positive or negative).

Modeling the above-mentioned learning preferences of Pre-classifier is not easy as it doesn't have a formal mathematical definition, but Machine Learning is essentially an optimization process that requires a definite objective. To address this problem, we use a sample weighted loss function as Eq.4.

$$\mathcal{L}(y_i, f_{\theta_j}(x)) = -\frac{1}{n} \sum w(s_i) \cdot \mathcal{L}_{\mathcal{CE}}(y_i, f_{\theta_j}(x_i)) \tag{4}$$

where $w(s_i)$ is the sample weight of $x_i$ based on the prediction $Score_i^{main}$ ($s_i$ for simplification in the formula) of Main-classifier.

## 3.3 WEIGHTED SAMPLING

To embody the above-mentioned learning preferences, following points are considered for the design of our sample weighting approach:

- For Positive samples, Pre-classifier is expected to focus more on the samples with high $Score^{main}$, for the reason that samples with low $Score^{main}$ are supposed to be rejected by Main-Classifier even they pass the Pre-classifier.

- For negative samples, it's more important to reject the ones with high $Score^{main}$ as it might be accepted by Main-classifier.

- If one sample's $Score^{main}$ is hugely high, it should be weighted higher, but not unlimited high.

- The weights for negative samples should have a lower bound in case too many of them are weighted near 0 as most of them are supposed to be scored very low by Main-classifier.

These principles can be formulated as in Eq.5 and curved in Figure 3.

$$w(s) = \begin{cases} min(\sigma(s - 0.5, t_{pos}, a_{pos}), w_{max}) & y = 1 \\ min(w_{neg\_min} + \sigma(s - 0.5, t_{neg}, a_{neg}), w_{max}) & y = 0 \end{cases} \tag{5}$$

$$\sigma(z, t, a) = \frac{a}{1 + exp(-\frac{z}{t})} \tag{6}$$

where $t$ is the temperature parameter, controlling the steep degree of the weighting function, $w_{max}$ and $w_{neg\_min}$ denote the upper bound of all samples and the lower bound of negative samples. Besides, $a$ is the attention intensity, controlling the theoretic fluctuation range of our weighting method. We force the model to pay more attention to the positive samples by setting $a_{pos} = 2$ and $a_{neg} = 1$.

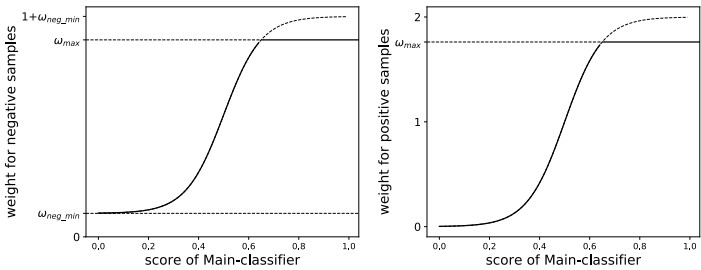

Figure 3: Sample weighting curves for negative and positive samples.

# 4 EXPERIMENTS AND RESULTS

## 4.1 EXPERIMENTAL SETUP

In this section, we describe the settings of our experiments.

**Datasets**

The proposed approach is empirically verified on two data sets.

1) **Task Extraction** is to judge whether one input sentence is a Commitment you made or a Request from others. We have training samples in 4 languages (PT, IT, FR, DE), each consists of 50K sentences machine translated from English. Evaluation samples are organic sentences from Enron emails Klimt & Yang (2004) and bad cases from user feedback, both annotated manually, 20K per language.

2) **Yahoo! Answers** Xiang Zhang & Lecun (2015) topic classification dataset has 10 main categories, each class contains 140K training samples and 6K testing samples. To take use of them in our binary classification scenario, we only consider class "Society & Culture" as positive while all others as negative. Only the "Answer" column is retained as the input of models. Besides, samples less than 5 tokens are filtered out. There are 1.46M (146K X 10) samples in total and they are split into 8:1:1 for training, validation and evaluation.

**Experiment Setup**

**Baselines:** we consider the Independent and Sequential Training as our baseline. Both frameworks share the same model architectures and settings with Feedback Training.

**Comparison Method:** Generally, the direct output of a binary classification model is a score, and we choose a threshold to decide whether it is positive or negative. In our two-stage Classifier, We define the term PassRate to denote the proportion of test samples passing through the Pre-classifier. For the sake of fairness, the PassRate for Pre-classifiers in Baseline and Treatments are set to be the same. In other words, they call the same times of Main-classifier and consumed the same computation resource.

**Implementation**

**Pre-classifier:** a Logistic Regression model trained by scikit-learn(0.24.1), using 30k token and Bi-gram features. The 30k features are selected using chi2-score from all tokens and Bi-gram features. Besides, balanced class weighting method King & Zeng (2001) is adopted to mitigate unbalanced distribution of class labels.

**Main-classifier:** a 12-layer Transformer-based model plus a softmax classification head, which has been pre-trained based on InfoXLM Chi et al. (2021), an XLM-Roberta Conneau et al. (2020) equivalent multi-lingual models. The fine-tuning is conducted by minimizing Cross-Entropy loss with Adam Kingma & Ba (2014) (lr = 0.00005 and batch-size = 64). It is a multi-lingual model which is shared by all languages when multiple languages are involved. The threshold of Main-classifier is 0.5.

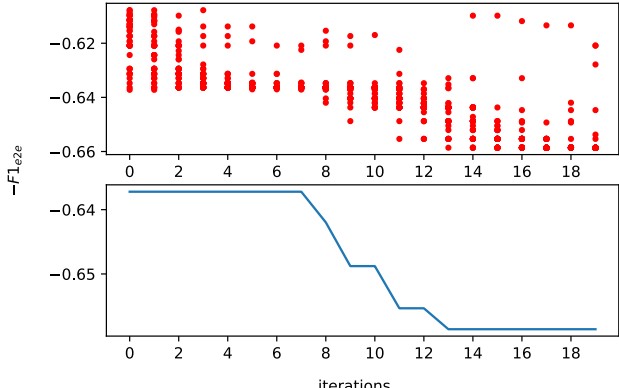

Figure 4: The optimization process of Feedback Training using Genetic Algorithm on Task Extraction Italian dataset.

| Lang. | $R_{pre}$ | $P_{main}$ | $R_{main}$ | $F1_{e2e}$ |
|---|---|---|---|---|
| | 0.6720 | 0.6988 | 0.8909 | 0.6449 |
| DE | 0.6720 | 0.7010 | 0.9004 | 0.6495 |
| | 0.6943 | 0.7214 | 0.9266 | **0.6801** |
| | 0.5946 | 0.5830 | 0.8977 | 0.5573 |
| FR | 0.5946 | 0.5681 | 0.8920 | 0.5292 |
| | 0.6047 | 0.5962 | 0.8994 | **0.5789** |
| | 0.6154 | 0.6538 | 0.9296 | 0.6102 |
| IT | 0.6154 | 0.6609 | 0.8984 | 0.6021 |
| | 0.6971 | 0.6585 | 0.9310 | **0.6537** |
| | 0.7168 | 0.6976 | 0.9259 | 0.6802 |
| PT | 0.7168 | 0.6958 | 0.9320 | 0.6817 |
| | 0.7301 | 0.7218 | 0.9272 | **0.7143** |

Table 1: Results on Task Extraction for different languages. In each group, the three lines denote the results of Independent, Sequential and Feedback Training. $P$ and $R$ denote precision and recall. Metrics footnoted with 'pre', 'main' and 'e2e' denote the performance of Pre-classifier, Main-classifier and the whole workflow.

**Genetic Algorithm (GA):** As the numerical optimization method is highly dependent on initialization and the underivable objective function is easy to be caught in abnormal solution. The $t_{pos}$, $t_{neg}$, $w_{neg\_min}$, $w_{max}$ are searched by Genetic Algorithm (using scikit-opt library) with the objective to maximize the $F1_{e2e}$ on validation set. Figure 4 shows an example of this optimization process for language Italian on Task Extraction dataset.

### Evaluation Metrics

We compare the performance of the multi-stage Classifiers in terms of F1, the geometric average of Precision (P) and Recall (R). The precision of the whole pipeline equals that of the Main-classifier in the final stage while the pipeline recall equals the production of that of all classifiers.

## 4.2    RESULTS AND ANALYSIS

**Task Extraction** We summarize the experimental results on the test set in 4 languages in Table 1, comparing the proposed Feedback Training framework with Independent and Sequential Training. For each language, the PassRate of Pre-classifier is the same for the control and treatment models. With Feedback Training, the e2e performances are consistently better than these of baseline in all 4 languages, demonstrating the proposed training strategy allows the whole pipeline to collaborate better than other training strategies. Sequential Training performs even worse than Independent Training for the reason that most of training samples are filtered out by Pre-classifier and the Main-model tends to overfit over the small number of the survived samples.

| PassRate | $R_{pre}$ | $P_{main}$ | $R_{main}$ | $F1_{e2e}$ |
|---|---|---|---|---|
| 20% | 0.6161 | 0.7059 | 0.5159 | 0.4383 |
|  | 0.6194 | 0.7244 | 0.5477 | 0.4462 |
| 30% | 0.7242 | 0.7009 | 0.4487 | 0.4440 |
|  | 0.7393 | 0.7167 | 0.4543 | 0.4574 |
| 40% | 0.8034 | 0.6987 | 0.4095 | 0.4473 |
|  | 0.7914 | 0.6923 | 0.4210 | 0.4499 |
| 50% | 0.8706 | 0.6952 | 0.3823 | 0.4502 |
|  | 0.8857 | 0.6932 | 0.3825 | 0.4551 |

Table 2: Experiments on Yahoo! Answer with different PassRates of Pre-classifier. In each group, the two lines denote the results of Independent and Feedback Training. The same goes for all the tables below.

| DataPortion | $P_{pre}$ | $R_{pre}$ | $P_{Main}$ | $R_{Main}$ | $R_{e2e}$ | $F1_{e2e}$ |
|---|---|---|---|---|---|---|
| 1% | 0.1604 | 0.4797 | 0.7128 | 0.4549 | 0.2182 | 0.3342 |
|  | 0.1666 | 0.6416 | 0.7174 | 0.4742 | 0.3042 | 0.4272 (+27.84%) |
| 2% | 0.1731 | 0.5158 | 0.7355 | 0.4742 | 0.2446 | 0.3671 |
|  | 0.1771 | 0.6668 | 0.7117 | 0.4856 | 0.3237 | 0.4450 (+21.23%) |
| 5% | 0.1904 | 0.5675 | 0.7210 | 0.4686 | 0.2661 | 0.3886 |
|  | 0.2244 | 0.6686 | 0.7075 | 0.4693 | 0.3138 | 0.4348 (+11.89%) |
| 10% | 0.2017 | 0.6011 | 0.7191 | 0.4607 | 0.2861 | 0.4094 |
|  | 0.2088 | 0.7179 | 0.7019 | 0.4665 | 0.3349 | 0.4535 (+10.77%) |

Table 3: Results of using different proportion of data to train Pre-classifer on Yahoo Answer dataset. The Pre-classifiers are trained with 1% to 10% training data while Main-classifiers is trained with full training set. PassRate of Pre-classifier is 30%.

**Yahoo! Answer** We further compare the proposed approach with Independent Training on this open-source dataset. Table 2 shows Feedback Training outperforms baseline in case of all PassRate of Pre-classifier. More notably, the smaller the PassRate, the greater the advantage. Examining the experiments with PassRate equals to 20% or 40%, although the recall of Pre-classifier remains unchanged under Feedback framework, the Main-classifier gets significant improvement. This demonstrates that Feedback method indeed learns to pass more samples that the Main-classifier can better deal with.

Experimental results under fewshot scenario are summarized in Table 3. Pre-classifiers are trained with 1% to 10% training data while Main-classifiers are trained with full training set. The Feedback Training shows superiority over the baseline, especially when less than 5% data is used to train Pre-classifier. The e2e gain of Feedback Training mainly derives from the much higher recall of Pre-classifier. Instead of focusing on the classification accuracy, the Pre-classifier is forced to pay more attention to recall more positive samples and meanwhile to avoid opposite forces from some hard negative samples that even can not be distinguished by Main-classifier.

| DataPortion | $P_{pre}$ | $R_{pre}$ | $P_{Main}$ | $R_{Main}$ | $R_{e2e}$ | $F1_{e2e}$ |
|---|---|---|---|---|---|---|
| 5% | 0.1904 | 0.5676 | 0.6676 | 0.4990 | 0.2832 | 0.3977 |
|  | 0.2007 | 0.7032 | 0.6560 | 0.5065 | 0.3562 | 0.4617 (+16.10%) |
| 10% | 0.2017 | 0.6011 | 0.6721 | 0.4854 | 0.2918 | 0.4069 |
|  | 0.2110 | 0.7148 | 0.6574 | 0.5011 | 0.3582 | 0.4637 (+13.96%) |
| 25% | 0.2191 | 0.6530 | 0.7290 | 0.4476 | 0.2923 | 0.4173 |
|  | 0.2162 | 0.7352 | 0.7175 | 0.4433 | 0.3259 | 0.4482 (+7.41%) |
| 50% | 0.2173 | 0.7180 | 0.7627 | 0.3984 | 0.2861 | 0.4161 |
|  | 0.2165 | 0.7348 | 0.760 | 0.3959 | 0.2909 | 0.4208 (+1.13%) |

Table 4: Results of using different proportion of data to train both Pre-classifer and Main-classifier on Yahoo Answer dataset.

More thorough fewshot experiments are conducted by using different proportion of data to train both Pre-classifier and Main-classifier, the results are shown in Table 4. The proposed approach still

shows its advantages for the following reasons: (1) Main-classifier trained with fewer samples tends to score higher to the test data (experimentally, as the input samples for Main-classifier are those passed Pre-classifier, and the Main-classifier here is worse than the one trained with full dataset. Hence it can not detect negative samples well and tend to give higher scores overall), thus acquires higher recall; (2) As the Main-classifier has higher recall, and the target of GA is to maximize the $F1_{e2e}$, therefore the learned Pre-classifier also prefers to acquire higher recall with the sacrifice of precision more intensively. With the combined effects of these two reasons, the power of Feedback Training is amplified. This result indicates the great value of Feedback Training when the training data is scarce.

## 5 CONCLUSIONS

This paper proposes a novel learning framework, Feedback Training, to improve the end-to-end performance of two-stage Classifier, the most common type of MSC. Experiments on both Task Extraction and Yahoo! Answer demonstrate the superior performance of the proposed approach and it becomes more advantageous when it comes to few-shot training scenario, reducing the requirements on the number of training samples for Pre-classifier in a great extent when a Main-classifier is given. The proposed Feedback Training method works only on two-stage binary-classification system, further work should be carried out to extend this approach to multi-stage multi-classification scenario.

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
