# OpenReview forum: "Training A Multi-stage Deep Classifier with Feedback Signals"
_ICLR.cc/2023/Conference — Submitted to ICLR 2023_

### Official Review · Reviewer_7ycA · 2022-10-21

**Confidence:** 2
**Correctness:** 3
**Technical Novelty And Significance:** 2
**Empirical Novelty And Significance:** 2
**Recommendation:** 1

**Clarity, Quality, Novelty And Reproducibility:**

Clarity: The paper is not well written and presented. The problem and the motivation of the proposed method should be well explained.

Quality: The quality of the paper is poor based on the presentation, technical novelty, and experimental evaluation.

Novelty: Both the problem and method are not new.

Reproducibility: There are not many details about the algorithm implementation. Therefore, the reproducibility is questionable.


**Strength And Weaknesses:**

Strength:

Developing an effective algorithm to trade-off between performance and model scale is an interesting research topic and practical for resource-constrained applications.


Weakness:

1. Figure 2 is not intuitive to explain the problem. And the motivation for proposing a reversed-order training framework is not well explained.

2. The logic of the proposed method is hard to follow. It’s not clear why the sample weighted loss function is able to address the challenge of modeling the learning preference as described in Sec. 3.2.

3. The weighted sampling strategy in Sec. 3.3 seems to be ad-hoc without any theoretical justification.

4. The experimental evaluation is weak without the comparison with some state-of-the-art comparisons.

5. Since the proposed method is only for the binary classifier, its practical usage is limited. It would be good to extend the method to multi-class classification.

6. There are works studying the efficiency of Committee-based models (ensembles or cascades), e.g. [a]. This paper lacks discussion and comparison of such related works.
[a] WISDOM OF COMMITTEES: AN OVERLOOKED APPROACH TO FASTER AND MORE ACCURATE MODELS, ICLR 2022


**Summary Of The Paper:**

This paper presents a training framework, named Feedback training, to improve the performance of two-stage binary classifier, a common type of multi-stage classification. Experiments on both task extraction and Yahoo Answer demonstrate the superior performance of the proposed method.

**Summary Of The Review:**

Given the comments about the weakness of the paper, the paper cannot meet the quality of ICLR.

---

### Official Review · Reviewer_WfBE · 2022-10-24

**Confidence:** 4
**Correctness:** 3
**Technical Novelty And Significance:** 2
**Empirical Novelty And Significance:** 2
**Recommendation:** 5

**Clarity, Quality, Novelty And Reproducibility:**

Clarity: fair
Quality: fair
Novelty: fair
Reproducibility: poor

**Strength And Weaknesses:**

Strengths.
This paper proposes a feedback training strategy, i.e., first train the main classifier, then determine the classification “difficulty” of each sample based on the trained main classifier, and finally adjust the weight of each sample during the training of the pre-classifier.
In general, the author’s motivation is clear, and proposes a simple yet effective (according to the experimental results presented in the paper) solution to solve the mentioned problems of previous Sequential Training and Independent Training methods.

Weaknesses
1) Although the paper is generally well organized and presented, there are also some obvious irregularities that may hinder understanding. For example, the meanings of “pre_score” and “main_score” in Figure 2 should be clearly expressed in the caption. For another example, for all tables in the experiment part, the corresponding training methods of each line should be indicated in the table rather than in the caption, and it is better to bold the best results for all tables for better clarity.
2) The reproducibility of the paper is doubted, since some details of the experiments are not specified. For example, for the few-shot experiment, 1%~10% of the training data is selected. However, how the data is selected and which data is selected (manually selected or randomly) are not clearly explained to ensure reproducibility. In addition, the specific value of PassRate for Table 1 is not specified.
3) Experiments can be more abundant and comprehensive to prove the effectiveness of the method. For example, more models can be adopted as the pre-classifier and the main-classifier. In addition, Figure 2 shows the inconsistency problem of independent training. I'd like to see a similar figure for feedback training, which can show more intuitively whether the pre-classifier achieves the claimed filtering effect. Further, for Yahoo! Answers dataset, the paper only splits "society&culture" into positive samples, and other classes are splitted as negative samples to meet the binary classification setting. I'd like to see the results of more splittings of classes to confirm that the results of baselines are consistently improved under different splittings.
4) Some minor grammatical errors need to be corrected. For example, the last paragraph of the first page: "exist methods" should be changed into "existing methods". The caption of Figure 1: "only samples passed" should be changed into "only samples that passed". The second paragraph of Section 3.1: "Transformer-base" should be changed into "Transformer-based", and "bias data set" into "biased data set". The second sentence of the first paragraph below Figure 2 ("but some other...") lacks a predicate. Experience Setup: Whether the first letter after the colon is in uppercase or lowercase should be determined.


**Summary Of The Paper:**

In view of the problems caused by Sequential Training and Independent Training in multi-stage classification, this paper proposes a new learning framework called feedback training, which outperforms the baseline methods in the two-stage binary-classification settings.

**Summary Of The Review:**

My main concerns are the writing, reproducibility, and significance of experiments.

---

### Official Review · Reviewer_ND4V · 2022-10-25

**Confidence:** 4
**Correctness:** 3
**Technical Novelty And Significance:** 2
**Empirical Novelty And Significance:** 2
**Recommendation:** 3

**Clarity, Quality, Novelty And Reproducibility:**

The idea to use multiple stage classifier is interesting, but I think it will be better to add more experiments to support the claims.

**Strength And Weaknesses:**

Quality/Clarity: the paper is well written and the techniques presented are ok to follow. The authors design a muti-stage classifier including main-classifier and the pre-classifier with the later is trained with weighted samples.

Originality/significance: the idea is incremental, there have been multiple existed approaches such cascaded classifier to speed up inference, adboost algorithm with weighted samples, etc.

Experiments: the method should be compared with some baselines such as adboost.


**Summary Of The Paper:**

This paper presents a novel training framework, named Feedback Training. The classifiers are trained in an reverse order, the main-classifier then followed by pre-classifier. And the pre-classifier at the later stage is trained with sample weighting method, which are based on the training result of Main-classifier. The experiments show the efficacy of the proposed approach, and its great advantage under the scenario of few-shot training.

**Summary Of The Review:**

Overall it is a good paper, but not ready for publication.

---

### Decision · Program_Chairs · 2023-01-20

**Decision:**

Reject

**Justification For Why Not Higher Score:**

Incremental novelty and insufficient experimental analysis. All three reviewers recommend rejection and there is no author response.

**Justification For Why Not Lower Score:**

N/A

**Metareview: Summary, Strengths And Weaknesses:**

The paper introduces a training framework for learning multi-stage classifiers with feedback signals. While the motivation is clear, all three reviewers agree that the idea is incremental and the experimental analysis is insufficient. There is no author response.

**Summary Of Ac-Reviewer Meeting:**

N/A